# Health technology assessment system in Switzerland: Current state, gaps, and prospects for improvement

Mohammed Alkhaldi[1,2,3,4,5]*, Rima Kachach[6], Line Enjalbert[2,3,4], Aisha Al Basuoni[7], Malak Alrubaie[8], Sara Ahmed[2,3,4]

**1** Department of Public Health, School of Health Sciences and Psychology, Canadian University Dubai, Dubai, United Arab Emirates, **2** Faculty of Medicine and Health Sciences, School of Physical and Occupational Therapy, McGill University, Montreal, Canada, **3** Centre for Outcomes Research and Evaluation (CORE), McGill University Health Center, Montreal, Canada, **4** Centre for Interdisciplinary Research in Rehabilitation of Greater Montreal (CRIR), The Integrated University Health and Social Services Centre of West-Central Montreal (CIUSSS West-Central Montreal), Center for Outcomes Research and Evaluation, Clinical Epidemiology, Montreal, Canada, **5** Centre for Tropical Medicine and Global Health, Nuffield Department of Medicine, University of Oxford, Oxford, United Kingdom, **6** Faculty of Health Sciences, The American University of Beirut, Beirut, Lebanon, **7** Projects Unit, Gaza Community Mental Health Programme, Gaza, Palestine, **8** Faculty of Health, University of Waterloo, Waterloo, Ontario, Canada

* mohammed.alkhaldi@mcgill.ca

## Abstract

Switzerland maintains a distinct HTA system that often aligns with EU requirements and reflects its unique healthcare governance model, with structured processes led by the Federal Office of Public Health (FOPH), despite lacking a comprehensive national framework. These processes, though sometimes limited in scope, produce nationally binding decisions. This study aims to comprehensively analyze and understand the key pillars of the national HTA system, identify obstacles, and propose insights to strengthen the HTA system in Switzerland. This system analysis study was conducted between 2021 and 2023, targeting key Swiss HTA-associated organizations and experts from various health sectors. Mixed methods were used to gather data using electronic institutional surveys completed by nine organizations to assess HTA technical aspects and virtual in-depth interviews (IDIs) with eight experts to understand HTA from a policy perspective. The study examined HTA practices in the Swiss organizations, comprising 67% academic, 22% public, and 11% non-governmental sectors. Findings revealed that 67% of the organizations mentioned that structured processes were led by the FOPH as the central agency (67%). While 56% (n = 5) of organizations reported substantial funding, challenges included a lack of government funding. In Switzerland, HTAs most frequently addressed safety, clinical effectiveness, and economic aspects, while social acceptability and feasibility were less commonly considered. Strong interest in expanding HTA capacity was expressed. However, there is limited awareness and advocacy for HTA, insufficient political support, limited institutional capacity, and persistent gaps in funding. The study highlights the importance

**Data availability statement:** The data contains identifying and potentially sensitive information related to participating organizations and experts. Data sharing is subject to restrictions imposed by the McGill Research Board Ethics Office in Canada. Access to the data may be requested from the corresponding author or the hosting institution upon reasonable request. Requests for data access can be directed to: The Faculty of Medicine and Health Sciences Research Ethics Board (McGill IRB), McGill University: Georgia Kalavritinos, Ethics Review Administrator, REB 1, 2, 3, georgia.kalavritinos@mcgill.ca.

**Funding:** This study was supported by McGill University, Montreal, Canada, in the form of a grant (MITACS – 259863, CHIRS – 259844) and by the Lindsay Foundation Grant at CIUSSS Centre-Sud-de-l'Ile-de-Montréal in the form of a grant awarded (CIUSS 2021). Both grants were awarded to M. Alkhaldi, the corresponding author of this study, whose specific roles are articulated in the "Author Contributions" section. The funders had no role in study design, data collection and analysis, decision to publish, or preparation of the manuscript.

**Competing interests:** The authors have declared that no competing interests exist.

of HTA in healthcare decision-making and resource allocation in Switzerland, and the need for addressing the integration gap of HTA into decision-making. Enhancing awareness and institutional strengthening are key to advancing HTA. This study offers insights to guide future research and to establish a national HTA agency, to improve integration and coordination of HTA across all Swiss Cantons.

## Introduction

Health technology assessment (HTA) is seen as a crucial element in any healthcare system based on solidarity, as it helps to inform funding decisions. Due to the growing discrepancy between an increasing demand for healthcare and limited resources, it is necessary to conduct a clear evaluation of how funds are allocated. Historically, HTA has relied on evidence-based medicine tools, such as a thorough review of top-quality research [1]. According to the World Health Organization (WHO), the quality and standard of healthcare have been improved through the use of HTA research evidence and information [2]. Applying this approach further advanced the Swiss healthcare industry due to the development of various healthcare tools and technologies, although the long-term consequences of their use are not fully understood. These consequences can be examined by HTA, which provides evidence that identifies both the advantages and disadvantages of all available options. In addition, the HTA process engages a range of stakeholders throughout and promotes channels for the dissemination of knowledge and evidence produced by HTA and supports greater transparency in the way healthcare decisions are made [3].

Several initiatives have been launched to improve communication and collaboration in the HTA field within Europe. EUnetHTA initially started in 2006 as a Framework European Union-funded project (2006–2008), followed by a one-year transition period, after which Joint Action 1 began in 2010. This Joint Action established EUnetHTA as a sustainable and effective HTA collaboration across Europe, designed to provide added value at the local, national, and EU levels. The EUnetHTA21 was composed of 35 government-appointed organizations from 24 EU member states, with additional participation from Switzerland, an EU member state, which is the country under the scope of this HTA system analysis study. Most EU countries developed their HTA programs organically, leading to significant variations in process and methodology. However, assessments typically summarize the available evidence to inform decisions on pricing or reimbursement, depending on the system [4]. All EU Member States have implemented HTA processes at the national or regional level in the past two decades [5]. Although there is some standardization in national HTA systems across Europe, notable differences remain in both process and methodology. HTAR (HTA Regulation, 2021/2282) is the new EU-wide framework that governs how Member States will jointly conduct clinical assessments of health technologies. While Switzerland is not part of the HTAR, its national regulations often align closely with EU requirements [6]. Switzerland has its own established HTA program, and this national program is distinct from the new EU system. The Swiss HTA landscape

reflects these differences in several ways, shaped by the country's unique healthcare governance model. With the HTAR, there is now a joint clinical assessment (JCA), while other domains, such as economic evaluations and decision-making processes, are handled differently across countries. It is important to note that all EU countries now follow the same process for clinical assessments, with joint assessments already in place for oncology and Advanced Therapy Medicinal Products, expanding soon to selected medical devices, orphan drugs in 2028, and all other therapies by 2030 [7].

A high level of complexity characterizes the Swiss healthcare system, blending elements of managed competition and corporatism within a decentralized regulatory framework influenced by direct democracy [8]. The statutory health insurance system and its benefit package in Switzerland are managed centrally at the federal level, with insurers providing coverage according to federal rules. This system contributes to Switzerland's high life expectancy (82.8 years), with healthy life expectancy exceeding the EU average [9].

According to the World Bank, Switzerland spent approximately 11.7% of its GDP on health expenditure in 2022 [10]. In Switzerland, the insurance and reimbursement system is notably fragmented and complex. The federal government centrally regulates statutory health insurance, including defining the mandatory benefit package, setting national reimbursement rules, and establishing uniform requirements for insurers. The cantons are responsible for implementing these federal regulations, financing and managing hospitals, and overseeing regional service delivery. The two federal and cantonal levels operate through a shared governance structure: federal legislation defines the scope of coverage, while the cantons ensure its practical application, resource allocation, and enforcement at the regional level. Mandatory Health Insurance with subsidies for low-income individuals ensures coverage [11]. The system offers a wide range of choices and direct access to all levels of care, with minimal waiting times, although insurance plans with gatekeeping restrictions are gaining prominence. Since 2000, the Swiss healthcare system has undergone reforms, including improvements to the MHI system, changes to hospital financing, stronger pharmaceutical regulations, epidemic control, and standardized human resources regulations across the country. The health insurance system in Switzerland is tax funded. In particular, the Swiss cantons cover more than 50% of inpatient costs, except for long-term nursing home care which has somewhat different rules. Individuals are required to purchase private health insurance, which contributes to the high out-of-pocket financial burden for households. However, there are also cantonal subsidies available for the health insurance premiums of lower-income families. The rising cost of medical technology and professionals is also a concern, as these costs are ultimately passed on to patients and the public who use the healthcare system [12]. Understanding the complexity and cost burden of the system is essential for conducting HTA studies, as they assess the value, effectiveness, and affordability of healthcare technologies while also navigating a governance environment where local autonomy can both facilitate tailored solutions and hinder standardization.

In Switzerland, the Coverage with Evidence Development (CED) approach for non-pharmaceutical technologies has been employed since 1996, which coincides with the Federal Law on Basic Health Insurance (KVG/LAMal) implementation. This law made it compulsory for every individual residing in the country to purchase a standard health insurance policy from one of roughly 60–70 rival health insurance providers [1]. The implementation of CED made it possible for people to access certain promising technologies in their early stages of development, and it may have prompted the creation of registries and research efforts, although many of these registries were of limited quality and utility [1]. However, the effects of these changes on patient outcomes and healthcare costs are unclear. Overall, HTA plays a significant role in shaping the Swiss healthcare system, ensuring that patients receive high-quality and cost-effective care while promoting transparency and accountability in decision-making. Switzerland would likely benefit from establishing a national HTA agency, similar to those in most other International Network of Agencies for HTA (INAHTA) member countries [13]. Yet, without a national HTA agency or strong inter-cantonal coordination mechanism, there is a risk of duplication and significant gaps in the assessment of health technologies.

In 2015, the federal authorities initiated an HTA program aimed at assessing currently reimbursed healthcare services. This program operates under the legal framework of Article 32 of the Federal Health Insurance Act (KVG; SR 832.10).

According to this article, medical services eligible for coverage by compulsory health insurance must satisfy efficacy, appropriateness, and cost-effectiveness criteria, which are subject to regular evaluation [14]. Conducting this study was essential to address a clear gap in the literature regarding HTA architecture, policy, and practice in Switzerland. Despite the country's strong commitment to evidence-informed healthcare, its HTA landscape remains fragmented, highly decentralized, and insufficiently assessed. To our knowledge, no prior research has systematically examined the key pillars of the HTA system, including stakeholder understanding, governance, policies, resources, capacity, and HTA implementation, through a system-analysis approach that enables a comprehensive understanding of national and canton-level dynamics. Generating this foundational evidence is crucial not only for informing effective knowledge translation strategies and supporting ongoing HTA reforms (including HTAR) at both the federal and local levels, but also for offering lessons for countries with similarly decentralized structures. Moreover, the insights derived from this study contribute to strengthening the evidence base needed to advance more robust HTA systems, develop effective policies, and promote best practices within Switzerland, while providing transferable learning for comparable health systems in the broader region.

## Research aims and objectives

The overall aim of this study was to comprehensively understand the main pillars of the HTA system in Switzerland and evaluate the current health technologies, services, and processes in the health system through the following specific objectives:

- Assess stakeholders' understanding of HTA concepts, their importance, and current practices;

- Examine HTA stewardship and governance structures, as well as capacities, resources, and the implementation and use of HTA in health decision-making and policymaking;

- Evaluate the extent to which current health technologies, services, and interventions are assessed using HTA;

- Identify gaps and propose feasible solutions, including a framework to support HTA best practices and knowledge translation strategies at national and regional levels.

## Methods

This national HTA system analysis was conducted between 4th October 2021–26th October 2023. The study used a cross-sectional design employing mixed methods, including an electronic institutional HTA survey informed by a literature review, and a virtual in-depth Interview (IDI), as outlined in the protocol and similar studies [15,16]. The HTA survey was administered to nine organizations involved in HTA to gather information on technical and operational aspects. The survey was completed by designated operational team members, such as HTA officers and staff, on behalf of their respective organizations. For the IDI, ten high-level experts from the same HTA organization representing various sectors were invited, and eight participated. The interviews focused primarily on policy-related aspects of HTA. The following section provides a structured discussion of both data collection tools.

## Ethics statement

The ethical approval from the McGill Ethics Institutional Board in Canada was obtained in 2021. However, this McGill ethical approval was renewed and obtained under the same REB File Numberin June 2025. The International Ethical Standards for Biomedical Research Involving Human Persons were followed for the implementation of this work [17,18]. Two consents were obtained; 1) institutional administrative consent (electronic via email) provided by heads/leaders of the Swiss participating organizations by accepting the invitation sent via email to participate in the survey, and 2) both individual consents, written via email and verbal via phone, were obtained where participants accepted the interview invitation sent via email/phone. Additionally, participants were asked to give their second verbal consent before the interview began.

These multiple consents were obtained to comply with ethical guidelines, ensure their voluntary participation, confidentiality, and the right to withdraw at any time, and their data would be discarded and not included in the final data set.

## Electronic Institutional HTA Survey

The HTA survey was adopted from the WHO global HTA survey [2]. This electronic survey consisted of six domains that cover the pillars of the HTA system, with each domain comprising relevant items (questions). These domains include understanding of HTA; the use and application of HTA; implementation of HTA; stewardship and management; resources and capacity; and impediments and insights for strengthening HTA. Further domains and questions of this survey on HTA processes, standardization, and HTA and decision-making were developed based on reviewing recent and relevant literature. The survey questions were closed-ended questions to collect data that reflects the technical, operational, and practical aspects of HTA. One survey was administered and completed by a member designated by the team in each of the nine organizations involved in HTA. The full survey is enclosed with Supplement 1.

## Virtual In-depth Interview

The IDI guide was developed according to best practices for qualitative research [19–21], ensuring a rigorous and systematic approach to data collection. Unlike surveys, which gather standardized responses, qualitative interviews enable an in-depth exploration of participants' policy perspectives, strategizing experiences, and decision-making processes. The IDI guide is enclosed in Supplement 2. The guide was designed using semi-structured interview techniques, providing a balance between consistency across interviews and flexibility to adapt to emergent themes. It aligns with established qualitative methodologies [22], and questions guided by the set of objectives and literature on HTA. Additionally, the guide adheres to recognized analysis and results reporting standards, such as COREQ (Consolidated Criteria for Reporting Qualitative Research) (enclosed in supplement 3) and SRQR (Standards for Reporting Qualitative Research), enhancing transparency, credibility, and analytical depth in HTA research. Both tools underwent a rigorous review and consultation process by ten recognized local and international experts in public health, health systems, digital technology, health economics, epidemiology, clinical specialties, and health policy and management. Feedback from these experts was incorporated into the final versions of the tools.

## Study population and sampling strategy

Maximum variation purposive sampling [23] was used for this study. The study identified two distinct groups of participants based on specific criteria. The first group included nine of the thirteen invited HTA-associated organizations operating in Switzerland's health sector, representing governmental, academic, private, and non-governmental (NGO) organizations. The management of these organizations internally assigned a technical/operational team, who runs all HTA processes to complete one institutional survey on behalf of their organization. The second group comprised ten experts/leaders from these organizations responsible for overseeing HTA policies and strategic issues. All were invited, and eight participated in individual IDIs.

Two methods were used to identify the names and relevant information on key organizations and individuals (experts and leaders)1) a rapid review of grey and published literature, and 2) extensive consultations with local collaborators to guide identification of organizations and experts in Switzerland. These methods helped to generate comprehensive lists of organizations for the survey and experts for the IDIs based on predefined criteria and conditions. Our target was to select up to ten existing local and national organizations and up to ten experts from the prepared list. After applying the inclusion criteria, thirteen major, active, and relevant HTA-associated organizations were selected that met most of the following conditions: the organization was officially recognized by local and national health authorities, active at least one year since its establishment, had a defined mission for HTA stewardship, production, education, research, and funding; had any previous or current programs, projects, or interventions that directly or indirectly related to HTA; or participated in one or more

 

of HTA activities. The organizations with no direct or indirect role in HTA and those that did not meet the inclusion criteria were excluded. A purposive sampling strategy, conducted in consultation with health authorities, was used to select the organizations that met the inclusion criteria. Thirteen selected organizations were invited to participate, and nine agreed. Their representatives were then asked to nominate or assign team members involved in technical and operational HTA processes within their organizations to complete the survey. Each team completed one survey on behalf of their respective organization. The inclusion criteria were also applied to the second group of experts, where ten experts responsible for and working in HTA policy, strategy, and systems were also identified from the same HTA-associated organizations in Switzerland. All ten experts were invited, and eight participated in the IDIs. Their roles are heads of HTA units, academics, policymakers, directors, and advisors. They had to be officially holding high-level positions in HTA or related to HTA at both local and national health systems, whether in HTA research, policy, management, or education. The experts with no direct or indirect role in HTA policy, strategy, and systems, and who did not meet the inclusion criteria, were excluded. Mixed sampling strategies were applied to determine the final and most relevant key informants and experts from the prepared lists. These sampling strategies included simple sampling, critical case, snowballing, convenience, and self-identified sampling [24]. These sampling strategies were also guided by the application of the pre-defined selection criteria that led to the selection of nine HTA-associated organizations and eight experts from the HTA community in Switzerland.

The research team, led by the Principal Investigator (MA), applied this selection approach to the recruitment of organizations and experts to ensure broad representation and consensus across sectors, organizations, and levels of research, policy, management, operations, and technical expertise. Given the limited size of the HTA community (i.e., organizations and experts working in HTA), the research team defined a sample size of five to ten organizations, with one individual recruited per organization.

## Data collection

The electronic HTA survey was developed using McGill's RedCap™ cloud-based clinical software. Each of the nine HTA-associated organizations completed one survey. This was executed by sending an official invitation outlining the research objectives to the head of each organization for approval. Once approved, the research team distributed the survey via e-mail to the nominated team leader of the organization to guide the technical, operational, practical, and managerial team/staff involved in HTA to complete the survey under the team leaders' guidance. The research team allowed all organizations and assigned teams a two-month period to complete and return the survey, and support to address any anticipated questions or issues related to the survey. The IDIs were conducted via a web audio-video conferencing platform (Zoom Communications Inc., 2020), and each IDI lasted between 45–60 minutes. Eight IDIs were conducted with eight experts from policy and strategy levels from different sectors, disciplines, and levels. The PI (MA) and research assistant (AAB) communicated with the selected experts and coordinated and conducted the virtual IDIs. The PI and research assistant attended the IDIs, the PI led the IDI and discussion, and the assistant facilitated and reported the IDI.

## Data management, analysis, and sample size

All data on organization names, experts, and data from the survey and IDI were stored on a secure McGill server and only accessible by authorized members of the research team. Survey and IDI data was stored on a secured server and then imported into two software programs for data management and analysis. Survey data was analysed using the IBM SPSS Statistics version 29 software program. The survey data were analysed using descriptive statistics, including frequency distribution, percentages, categories, means, and standard deviation. Comparisons were made between organizations and sectors. IDI data was audio-recorded and then simultaneously translated and transcribed in English into MS Word sheets by the PI, assisted by trained co-investigators. Transcripts were imported into the software program, MAXQDA 12 (VERBI GmbH, Berlin), for qualitative data management and analysis. Transcripts were checked by the PI to ensure quality. Two coders and co-investigators constructed and validated codes in MAXQDA by classifying transcripts into IDI using

a preset coding system that was derived from study objectives. To maintain consistency, a third independent reviewer resolved disagreements. Peer iterative review of themes, participant feedback, and triangulation with survey data were performed by the PI and reviewers to strengthen credibility. The methodological approach of this study was informed by similar studies [20,21,16,25,26]. The COREQ approach was followed for reporting the study results. The IDI transcripts were analyzed using thematic analysis, guided by both deductive and grounded theory approaches. The research team used a study framework, developed through expert consultation and literature review, based on six HTA system pillars to ensure systematic and rigorous interpretation. The sample size was guided by a maximum variation and determined based on availability and accessibility, relevance, diversity, and representation of the essential organizations and experts involved in HTA, taking into account the practical, operational, technical, policy, and scientific considerations raised from the consultations mentioned above.

## Results

### Survey Findings on HTA System from a Technical Perspective

Nine organizations in Switzerland completed the survey. The majority, 67% (n = 6), from academic organizations, 22% (n = 2) from the public sector (governmental), and 11% (n = 1) from the non-governmental sector. The majority of organizations were employed in medicine (67%, n = 6) and multidisciplinary fields (44%, n = 4), while 11% (n = 1) reported working in 11% (n = 1) in economics.

All nine surveyed organizations demonstrated a high level of understanding of HTA purpose and HTA concept, with 89% (n = 8) reporting a high level and 11% (n = 1) a moderate level of understanding, respectively. The findings also indicated that a good level of HTA application was reported within these organizations, as well as a strong recognition of HTA's importance in their respective areas of work (67%, n = 6). Six organizations (67%, n = 6) indicated that the Swiss Federal Office of Public Health (12) was the central agency responsible for HTA management. Two organizations (22%, n = 2) indicated there was no central agency responsible for HTA management, and 11% (n = 1) indicated they did not know. Five organizations (56%, n = 5) reported they had a legislative requirement to consider the process and results of HTAs in the financing and public health decision-making process, while 22% (n = 2) reported they did not or did not know, respectively.

Concerning the HTA reports, only 44% (n = 4) of organizations reported that a national agency unit or committee produced HTA reports, while 44% (n = 4) reported they had not, and 11% (n = 1) that they did not know. Organizations mainly chose the Ministry of Health as the main recipient of the HTA reports (89%, n = 8), and at the same time minority of organizations chose also HTA reports were also intended for a National Independent committee related to HTA (22%, n = 2) and clinician associations (11%, n = 1) (Fig 1). All organizations indicated that publication and dissemination of HTA conclusions and reports occurred through organization websites (56%, n = 5) or public online platforms (44%, n = 4). Almost half of the surveyed organizations (56%) reported that civil society can give feedback on recommendations of an HTA report, and 67% (n = 6) also declared that stakeholders, including the community, can review a draft version of the assessment before the report is finalized.

In Fig 2, organizations highlighted that most of the HTA professionals were involved in various HTA processes: topic selection, scoping, systematic literature search, evidence collection, synthesis, modeling, and review, and they were mainly from public health and clinical science. Also, 88% (n = 8) of the organizations in Switzerland considered that HTA evaluations were conducted by other organizations or countries such as Germany, Austria, Haute Autorité de Santé (HAS) in France, NICE-UK, and Swiss Medical Board.

Regarding capacity support, only 11% (n = 1) of Swiss organizations reported measuring the impact of HTA on decision-making. In the 12 months preceding the survey, the Swiss Ministry of Health was the most frequent requester of assessments, either almost always (50%, n = 4) or frequently (25%, n = 2). Public healthcare providers requested assessments almost always (40%, n = 4) or sometimes (20%, n = 2). Similarly, public insurance bodies, reimbursement agencies, or national coverage institutions requested assessments almost always (33.3%, n = 3) or frequently (33.3%, n = 3) (Fig. 3).

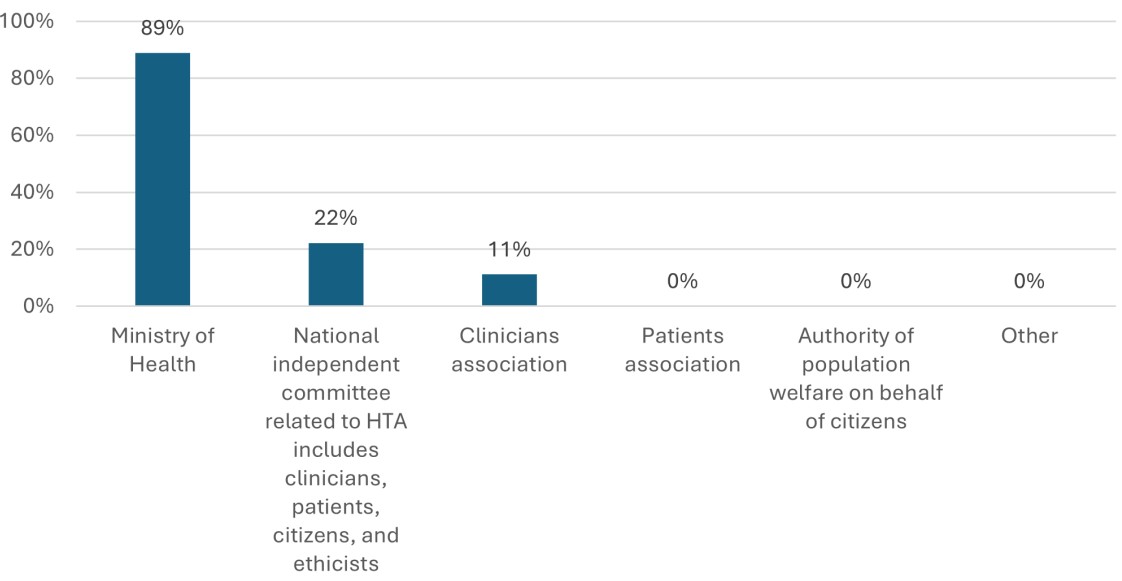

**Fig 1. Recipients of the HTA reports.**

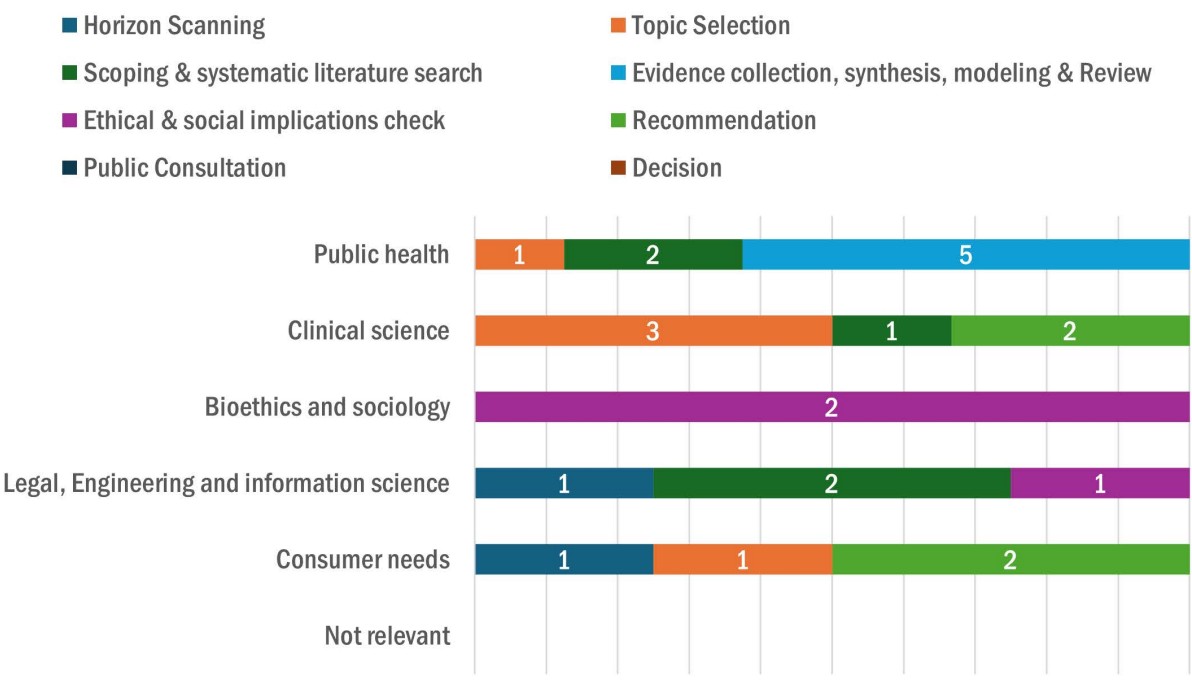

**Fig 2. Relevant stage or step of the HTA process, the following professional human resources.**

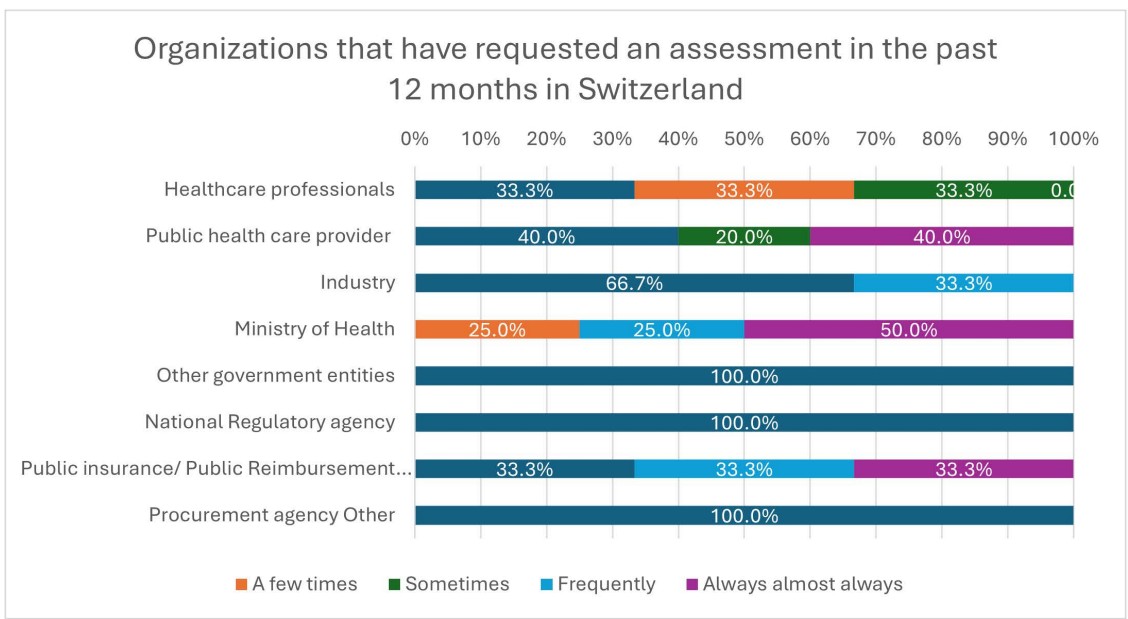

**Fig 3. Organizations that have requested an assessment in the past 12 months in Switzerland.**

All organizations estimated that less than 20 professional staff were involved in the HTA unit, agency, or committee: 60% (n = 5) estimated a number between 6 and 20 professionals, and 40% (n = 4) between 1 and 5 professionals. Fig 4 shows that the majority of professional human resources contributing to the preparation of HTA reports were involved in clinical interventions, including mainly professionals from public health and clinical science, as well as legal, engineering, and information science.

Fig 5 presents the most important dimensions of value for HTA that meet organizations' or Switzerland's health priorities and needs. The most important dimensions of value were clinical effectiveness and costs and economic evaluation (89% each, n = 8). Dimensions of safety, equity, and equality issues came in second position with 44%, and feasibility considerations and ethical issues with 33% (n = 3).

Among all organizations, 67% (n = 6) indicated that assessments were most likely conducted for new technologies, while 56% (n = 5) noted the same for established or widely adopted practices. In contrast, only 22% (n = 2) reported that assessments addressed emerging technologies, further developments of existing technologies, or those in declining use. The frequency of covering different aspects of HTA by type of technology in Switzerland is presented in Fig 6. Safety, clinical effectiveness, costs, and economic evaluation were the most common aspects covered in HTA in all types of technologies. Patients/citizens'/community' acceptability, views, communication and involvement, and feasibility considerations were the least common aspects covered in HTA.

Two-thirds of organizations (67%, n = 6) declared they used guidelines, models, frameworks, manuals, toolkits, or standards for preparing and developing HTA in their institutions in Switzerland, including 45% (n = 4) from EUnetHTA guidelines, 11% (n = 1) from WHO guidelines, and 11% (n = 1) from others.

Fig 7 shows the different areas where guidelines were applied/utilized for producing and preparing HTA reports, and where technical practices or procedures were applied for submitting HTA reports and mechanisms for communicating them. Guidelines were mainly used in medicines (78%, n = 7), followed by clinical interventions (67%, n = 6) and medical devices (56%, n = 5). Technical practices were also applied/utilized for clinical interventions (56%, n = 5), followed by medical devices and medicines (44.4% each, n = 4).

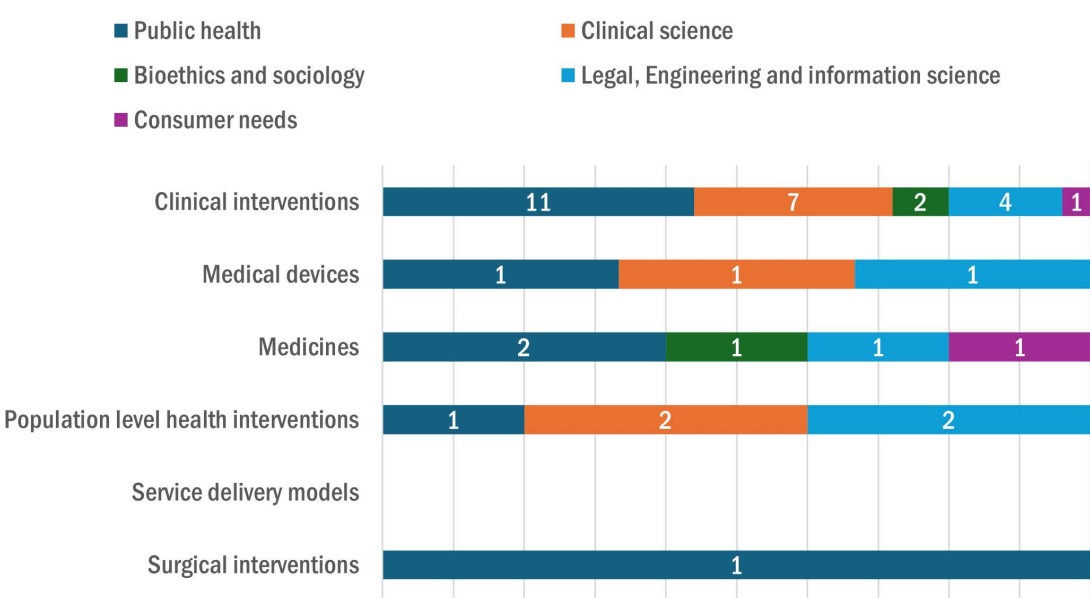

**Fig 4. Professional Human Resources' contribution to the preparation of HTA reports.**

Timelines are very important in HTA processes, and Fig 8 shows the steps of the HTA process for which timelines did or did not exist, whether they were transparent or whether they were binding. For HTA conduction and HTA submission, 71.4% (n = 5) of organizations declared that transparent and binding timelines existed. Half of the organizations declared that timelines were not transparent for the appeal procedure and that no timeline existed for the decision process.

All organizations declared they have a formal process by which information was gathered to support decision-making on new devices, drugs, or vaccines. Organizations revealed that HTA was mainly used in Switzerland for reimbursement decisions (56%, n = 5) and support of clinical guidance (44% each, n = 4), as well as coverage (33%, n = 3). Some organizations also reported that HTA was initiated by manufacturers, authorities, or academia (22% each, n = 2) or by others (34%, n = 3), including hospitals or combinations of the previous answer. Only two organizations declared that an HTA question or area had been identified by a prioritization exercise or scientific advice and recommendation (25% each, n = 2), while 12% (n = 1) declared that it was by a decision from authorities, and the last 38% (n = 3) by some other method.

In Switzerland, 45% (n = 4) of organizations considered their information-gathering practices on new interventions as part of HTA, while 33% (n = 3) did not, and 22% (n = 2) were unsure. The main purpose of gathering information is to inform reimbursement and benefits package decisions (78%, n = 7) and the pricing of health products (56%, n = 5) (Fig 9).

In Switzerland, half of the organizations (50%, n = 4) relied primarily on secondary sources (e.g., databases, systematic reviews, gray literature) for HTA, while 38% (n = 3) used both primary (e.g., registries, observational studies) and secondary sources; one organization (12%, n = 1) was unsure. Only 38% (n = 3) of organizations declared that HTA data was collected by authorities, while 12% (n = 1) reported that it was submitted by a manufacturer and 50% (n = 4) by another method. The most important areas where HTA was used as an element of measurement in the decision-making process

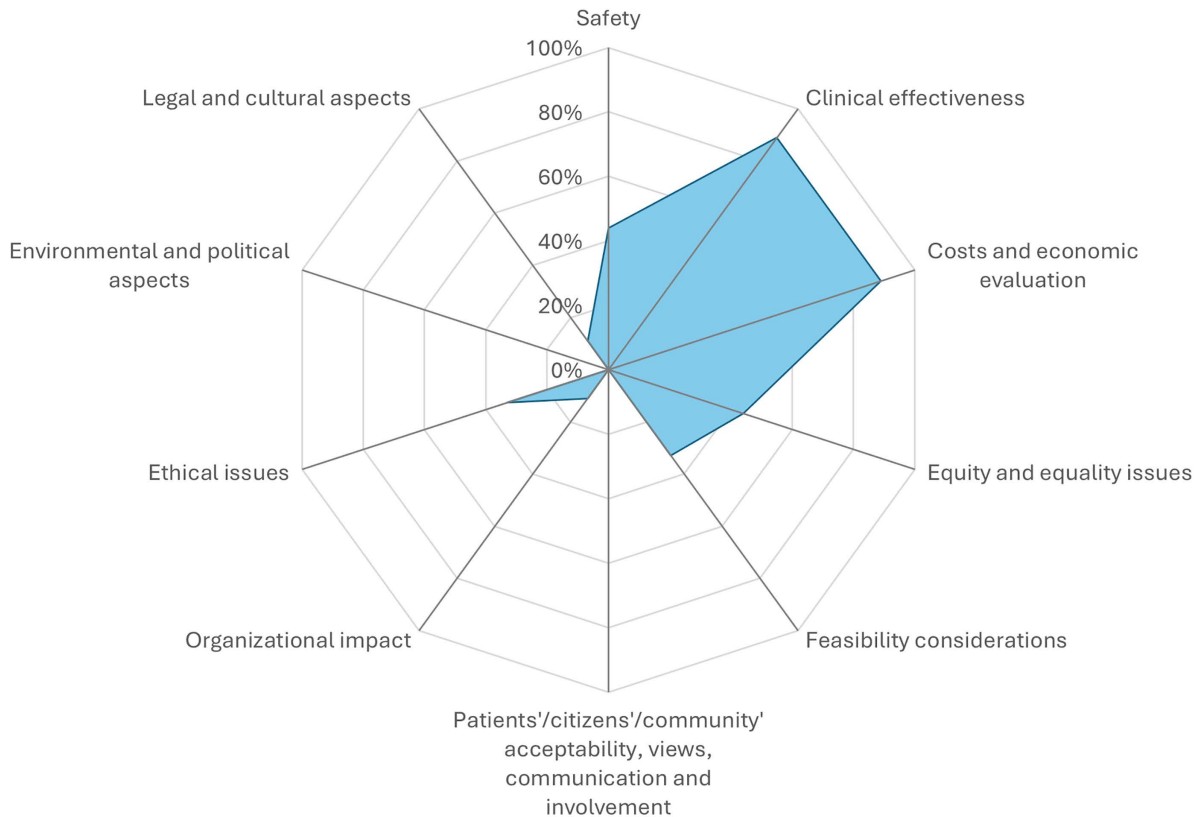

Proportion of most important dimensions of value for HTA that meet the institution/country's health priorities and needs

**Fig 5. Proportion of the most important dimensions of value for HTA.**

were clinical interventions and medical devices (89% each, n=8), medicines (78%, n=7), surgical interventions (67%, n=6), and followed by population-level health interventions (56%, n=5), vaccines (33%, n=3) and digital technologies (11%, n=1) (Fig 10).

In Switzerland, reports were used on a national basis by legislation for 38% (n=3) of the organizations, on an institutional entity basis by 25% (n=2), and both for 25% (n=2). For half of the organizations, HTA is only one element of informing the decision, whereas the organization making the decision relied partially (25%, n=2) or completely (12%, n=1) on the conclusions of the assessment.

Fig 11 presents the most significant impediments at the country level to the use of HTA in health care policy decision-making. As reported by 56% (n=5) of the organizations, the principal impediments were the lack of awareness and advocacy of the importance of HTA, the political support, and the institutional capacity, such as the setting-up process and well-functioning operations. The lack of qualified human resources (44%, n=4) and institutionalization of HTA (33%, n=3) were also noted. To strengthen HTA capabilities and production structure, the organizations highlighted actions as very important, such as awareness of HTA advantages and institutional strengthening (62.5% each, n=6), and as fairly important as information availability (63%, n=6) and budget increase (50%, n=5), as illustrated in Fig 12.

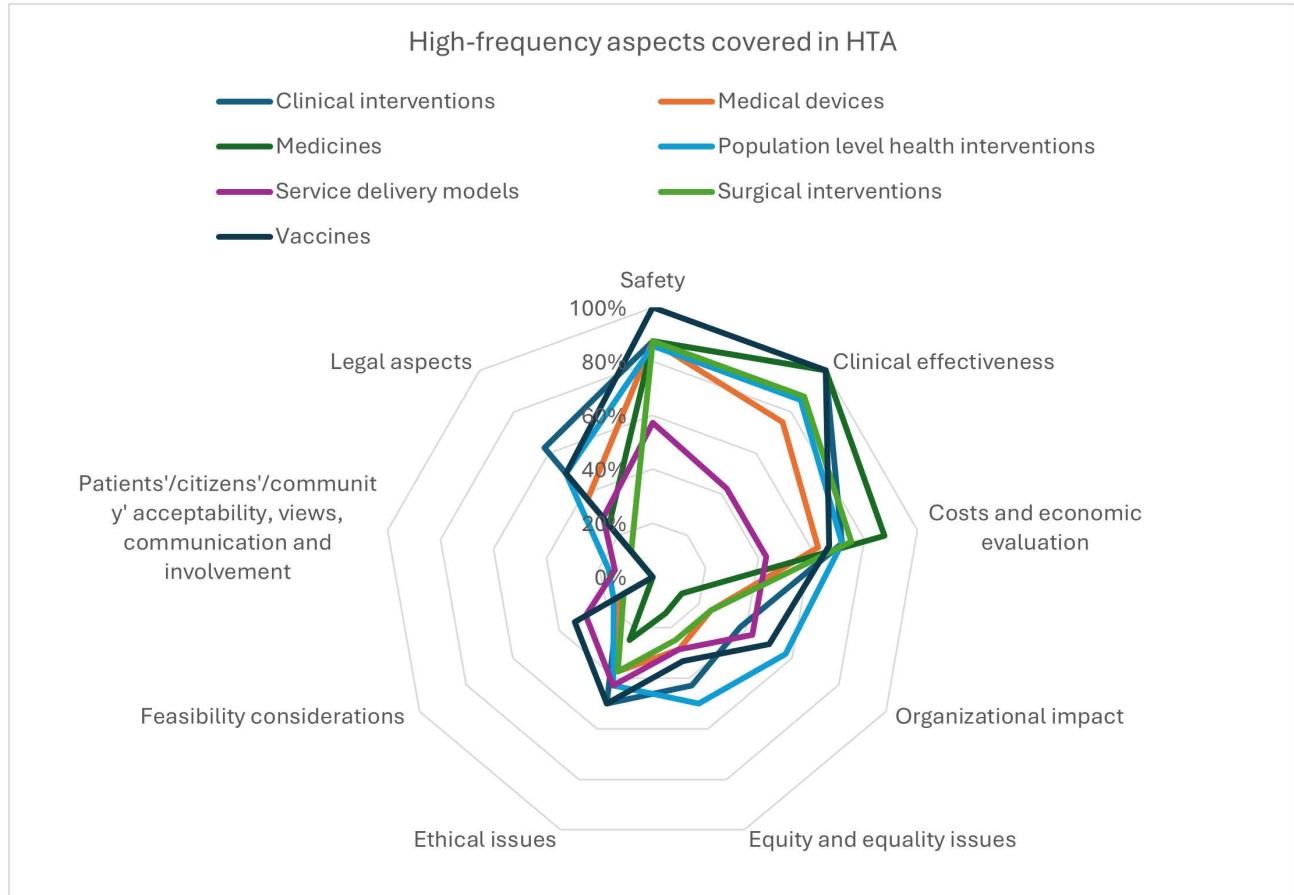

**Fig 6. High-frequency aspects covered in HTA in Switzerland.**

The final section of this system analysis focuses on the academic or training programs supporting HTA capacity building in Switzerland. Findings reported (56%, n = 5) that these HTA programs are introduced and conducted through courses, seminars, or workshops. Fig 13 shows some interest by the Swiss organizations in international HTA training and knowledge platforms for enabling the continuous education on HTA in Switzerland.

## Qualitative Evidence on HTA System: Policy Perspectives

A total of eight experts involved in HTA policies and strategies from various organizations across Switzerland attended IDIs. Of these experts, one was from the private sector, one from NGOs, and six were from academic organizations.

## Understanding and Perceptions of HTA Application

Experts demonstrated a strong understanding of the purpose and concept of HTA. Those from academic backgrounds, in particular, provided a comprehensive view of HTA, emphasizing its role as both a systematic evaluation process and a multidisciplinary endeavor. Several described HTA as "systematic and interdisciplinary scientific research," with academic-affiliated experts.

Several experts also emphasized the significant impact HTA has on health outcomes and policy, noting its importance in ensuring that healthcare interventions are both effective and efficient. There was broad agreement that HTA is essential

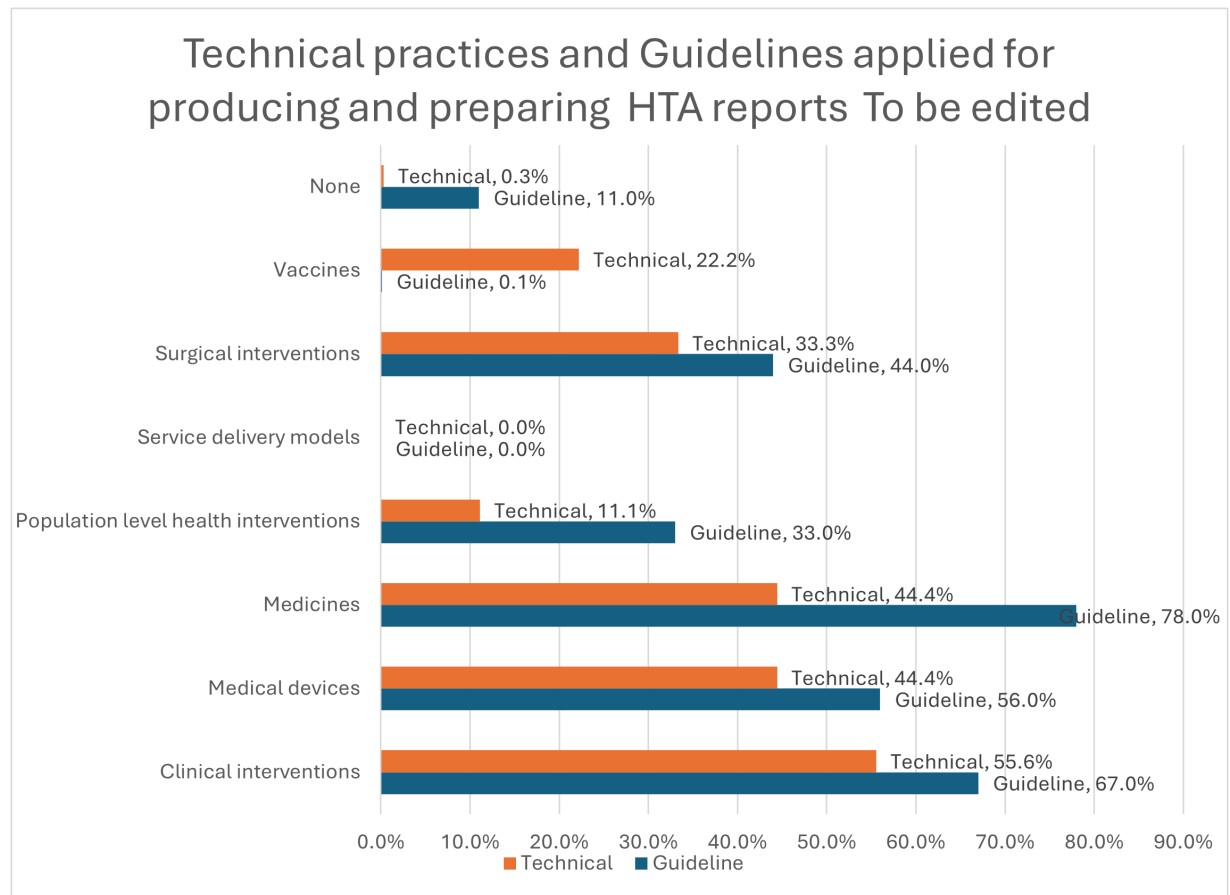

**Fig 7. Technical practices and Guidelines applied for producing and preparing HTA reports.**

for rational decision-making in healthcare and HTA was viewed as a critical tool for improving transparency and account-ability in health policy as well as for fostering evidence-based practice. However, some acknowledge that HTA benefits can vary depending on how well it is understood and applied across different sectors.

**Stewardship and management**

Most experts identified the FOPH as the principal regulatory body overseeing HTA in Switzerland, while also acknowledg-ing the involvement of academia, insurance firms, health providers, and patients in the HTA process. However, there were divergent views on the existence of national HTA policies, with three experts noting the absence of a HTA national policy and one academician mentioning an initiative launched 12 years ago: *"So the Swiss Academy of Medical Sciences and the conglomerate of all the cantons decided that we wanted to have HTA. It was, as I said about 12 years ago…"* (Inter-viewee S6, Academic Expert).

Experts indicated that most professionals engaged in the HTA process came from clinical interventions and pub-lic health or clinical science backgrounds, with noticeably less involvement from professionals specializing in medical devices, medicines, or population-level health interventions. While the multidisciplinary nature of HTA was recognized, expertise was seen as concentrated in a few key areas.

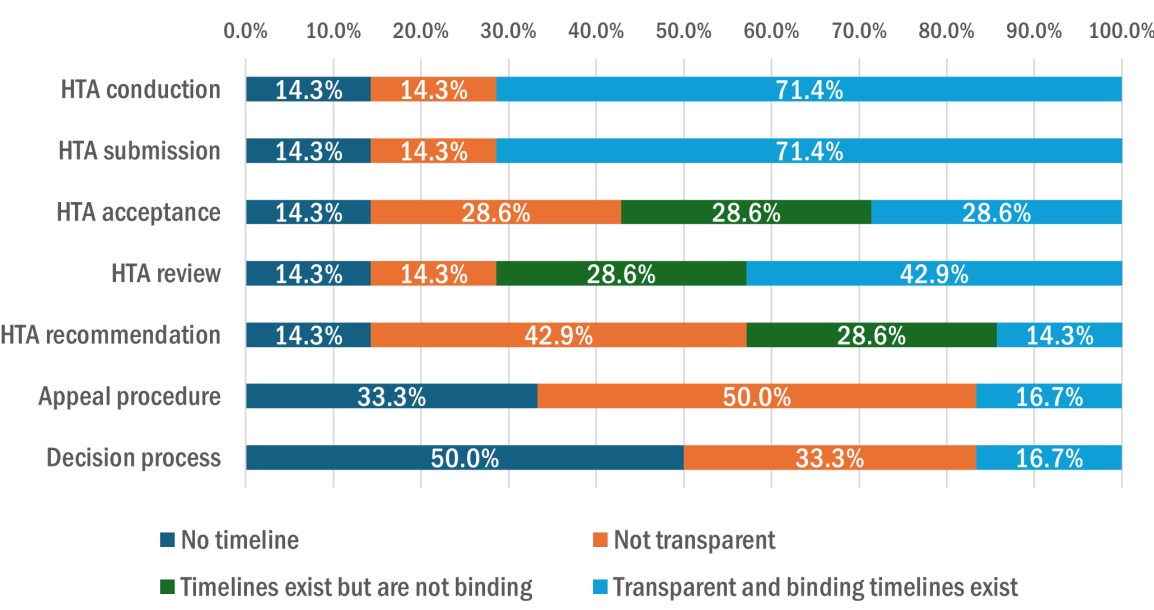
Fig 8. Timelines of the HTA process in Switzerland.

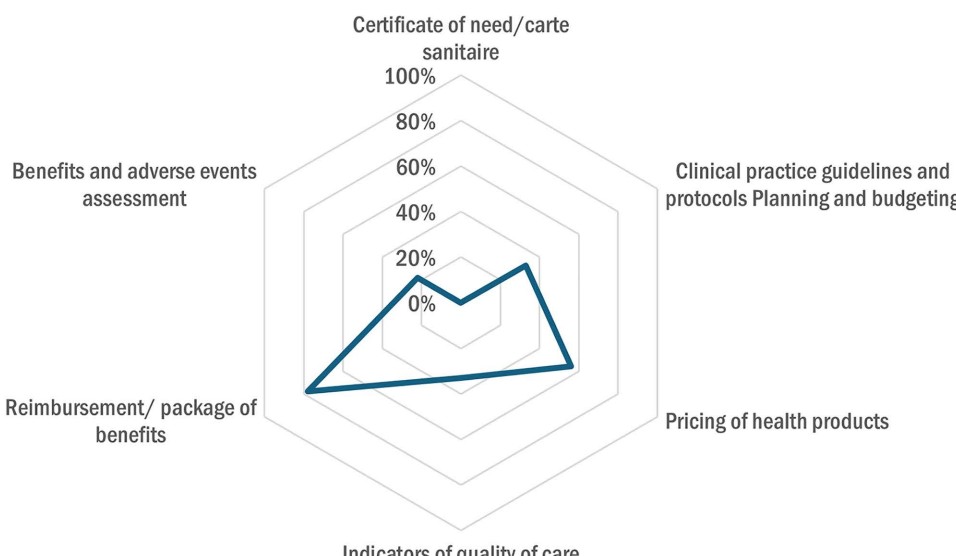

Fig 9. Purpose of undertaking HTA in Swiss organizations.

Regarding HTA reports, experts expressed dissatisfaction with communication, coordination, and the practical use of HTA outputs. As one expert stated: "I'm not really satisfied. The Swiss Medical Board model was satisfying to quite an extent—it produced good, high-quality outputs and good dissemination—but it had no decision power, so it was poorly

Areas where HTA is used as an element of  measurements the decision-making process

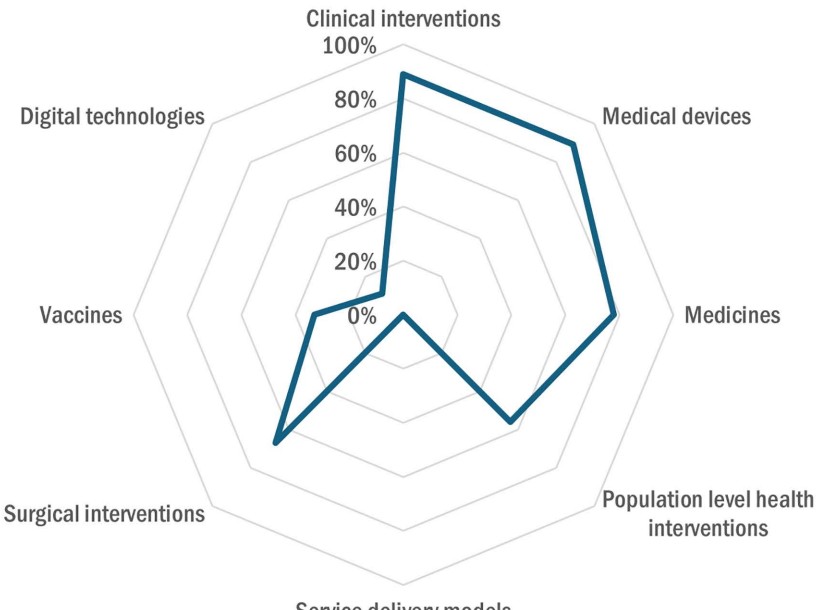

**Fig 10.  Areas where HTA is used as an element of measurement in the decision-making process.**

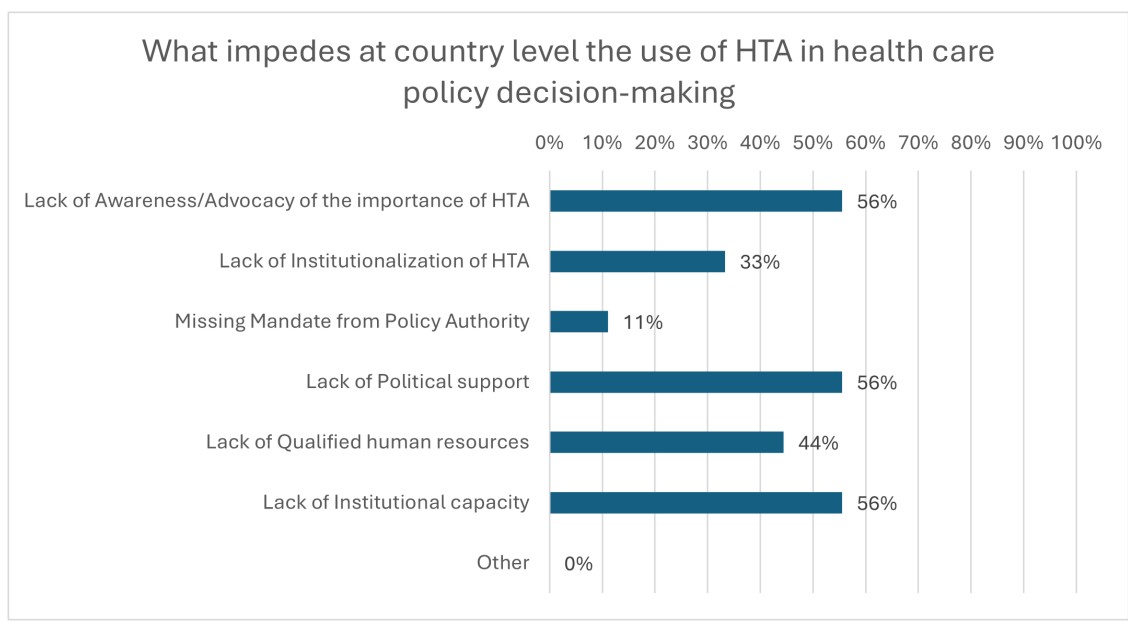

**Fig 11.  Country-Level Impediments to the Use of HTA in Healthcare Policy Decision-Making.**

## What would help to strengthen HTA production capabilities and structure

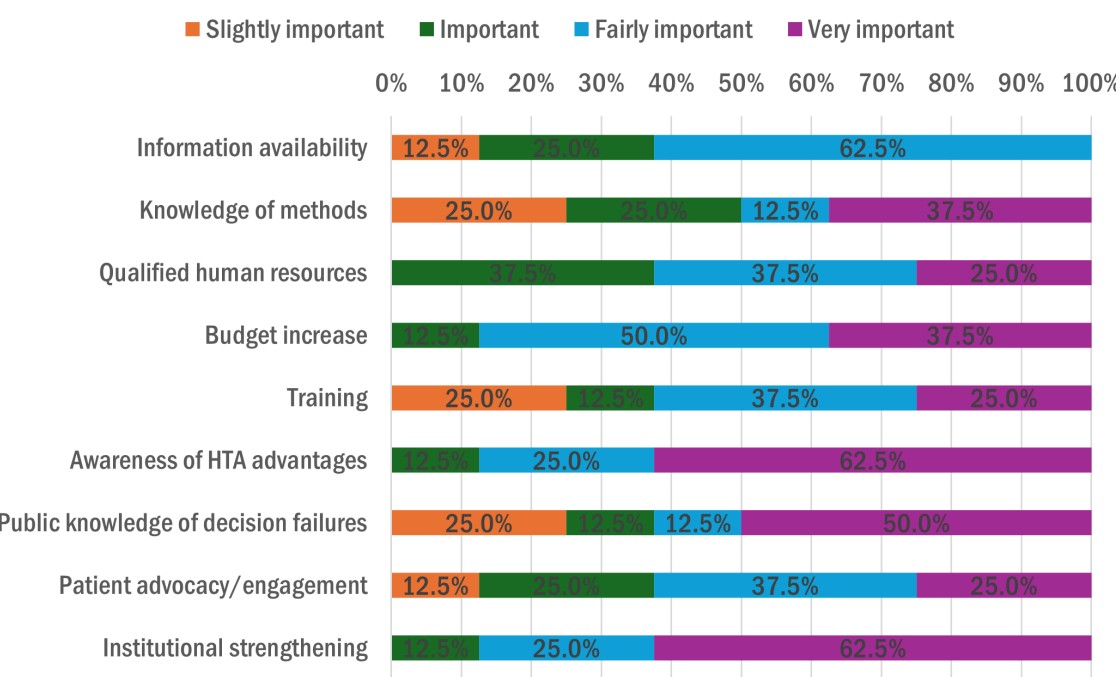

**Fig 12. Factors that would help to strengthen HTA production capabilities and structure.**

## Interest in international HTA training and knowledge platform for continuous education on HTA

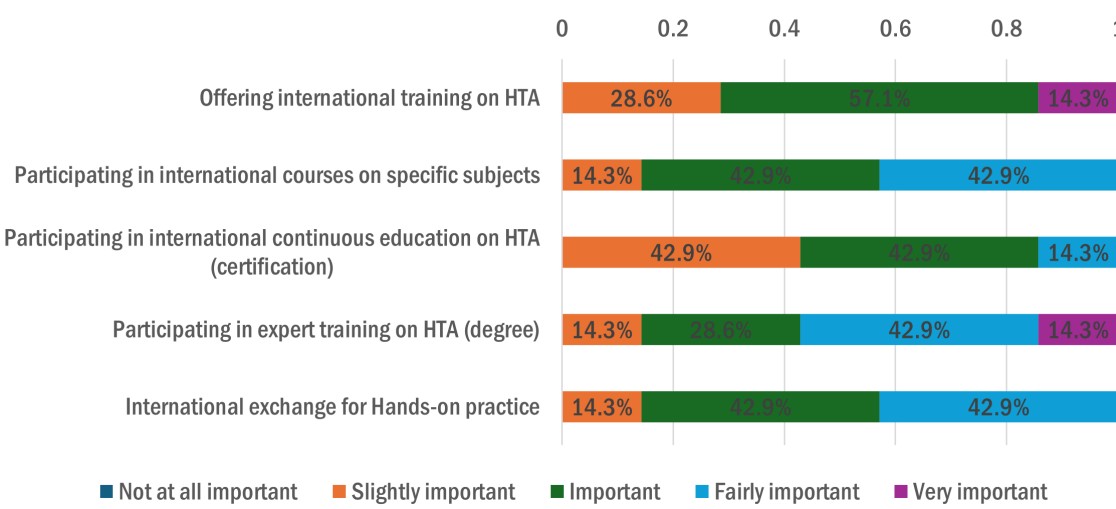

**Fig 13. Interest in International HTA training and knowledge.**

utilized" (Interviewee S2, Academic Expert). Conversely, a different expert noted positive trends in engagement and evidence use: *"So I'm satisfied that there is a growing interest from clinicians and other actors in the system to get economic evidence, so that's a positive trend"* (Interviewee S3, Academic Expert).

## Resources and Capacity Supporting the HTA Process

Experts consistently highlighted a significant lack of dedicated government funding for HTA in Switzerland, and many noted that costs are often covered by external stakeholders, such as private companies, rather than public authorities. One expert highlighted this challenge and said, *"There is a lack of funds that go into HTA from the government or public authority sites; there needs to be funding for more people who can be trained"* (Interviewee S7, Academic Expert).

Beyond the challenge of limited funding, experts also highlighted the need for multidisciplinary expertise and additional trained personnel as major barriers to expanding HTA capacity.

## Implementation process of HTA

Clinical effectiveness and economic evaluation were identified as the most important value dimensions for HTA. Safety, equity, and ethical issues were mentioned less frequently, while patient/community acceptability, communication, involvement, and environmental or political aspects were rarely prioritized. HTA assessments were reported to be more commonly conducted on new technologies or established practices, with less attention given to emerging or declining technologies.

Guidelines and standards from international organizations (such as EUnetHTA, WHO, INAHTA, and ISPOR) were acknowledged as important references for HTA guidelines, especially in the context of medicines and clinical interventions. However, the application of such guidelines was described as variable, depending on the specific context and resources available.

## Use and application of HTA in decision-making

Experts described HTA as being used primarily to inform reimbursement decisions (i.e., determining payment levels for covered medicines, services, technologies, or products), to support clinical guidance, and, to a lesser extent, to inform coverage decisions (i.e., whether technology is included in the health system). The initiation of HTA assessments varied, with manufacturers, authorities, academia, and hospitals all playing key roles, reflecting a collaborative and sometimes fragmented approach. Information gathered through HTA was primarily used for decisions on reimbursement, benefit package design, and health product pricing. HTA was most frequently applied to clinical interventions, medical devices, and medicines, with less common use in surgical and population-level interventions.

While experts agreed that HTA supports rational healthcare decision-making, they noted that its influence on policy is often limited, as political considerations frequently outweigh HTA recommendations. As one academician put it, *"HTA is part of decision making but political aspects play a bigger role"* (Interviewee S8, Academic Expert).

## Interests and impediments to building capacity

Strong interest in expanding HTA capacity in Switzerland was expressed, recognizing its value for evidence-based decision-making. However, significant impediments were consistently pointed out, including a lack of awareness and advocacy for HTA, insufficient political support, limited institutional capacity, and the persistent funding and training gap. Experts stressed the need for more multidisciplinary expertise and qualified personnel, noting that most current HTA professionals come from clinical and public health backgrounds, with fewer from other relevant fields. While academic and training programs exist in Switzerland, such as courses, seminars, and workshops, these were seen as insufficient to meet the growing demand for HTA skills.

## Discussion

HTA serves as a critical tool for evaluating how health resources, infrastructure, technologies, and funding are utilized, relying on rigorous, evidence-based reviews of high-quality research [1]. While HTA processes have been implemented across much of Europe, Switzerland's healthcare landscape is somewhat inconsistent, reflecting complex regulations and a highly decentralized system [8]. In this context, HTA plays a vital role in promoting high-quality, cost-effective care and ensuring transparency in healthcare decision-making. However, the substantial autonomy of cantonal authorities influences how policies are interpreted and applied, creating unique challenges and opportunities for the implementation and harmonization of HTA practices within the country.

This study provides critical insights into Switzerland's HTA landscape by examining key pillars of the system, including stakeholder understanding, governance, resources and capacity, HTA implementation and utilization in decision-making, evaluation of health technologies, and identifying gaps to propose solutions for best practices and knowledge translation, as well as assessing the value, effectiveness, and affordability of healthcare technologies in the country. To the best of our knowledge, this system analysis considers the first national study conducted on HTA in Switzerland. The study highlighted key findings related to these pillars of the HTA system.

Regarding the understanding of HTA's purpose and concept, the findings revealed a high level of awareness, with respondents broadly recognizing its applicability. Such awareness underscores the recognition of HTA's value in supporting evidence-based decision-making. Additionally, respondents also highlighted its applicability and importance within their respective organizations, particularly in clinical guidance and guiding reimbursement decisions. This suggests that HTA is not merely a theoretical or academic exercise, but is also regarded as integral to the daily and strategic work of healthcare leadership. Regarding HTA governance, despite a strong overall understanding of HTA, inconsistencies remain in how it is implemented across organizations and regions, pointing to persistent challenges around standardization and coordination that appear closely linked to Switzerland's high degree of decentralization. While this structure enables adaptability to local needs, it can also result in uneven adoption of innovations, duplication of assessments, and delays in achieving consistent national practice. Further regarding HTA governance, the process for collecting HTA information appears to be well-organized and structured in a significant proportion of organizations and is frequently used to aid decision-making in Switzerland. The central role of the FOPH in this process cannot be overstated, with an HTA unit being established within the FOPH in 2017 following a resolution by the Federal Council in 2015 to enhance HTA efforts [14,27]. This governance structure aims to bridge the gap between research and policymaking by facilitating the collection and dissemination of HTA information. The resulting HTA reports that enhance transparency by enabling civil society and stakeholders to review findings [14]. However, the absence of clarity among respondents regarding legislative requirements for incorporating HTA results into financing and public health decision-making remains. It is evident that there is a strong need for clearer communication and more robust policy frameworks, as some participants were uncertain or unaware that such policies existed.

For the HTA resources and capacity, our findings highlight that HTA activities in Switzerland are primarily supported by government funding, with the private sector playing a modest supplementary role. While the current reliance on government funding model provides a foundation level of stability, respondents identified funding gaps and relative limited number of professional staff dedicated to HTA at the national as a significant barrier to expanding HTA capacity. This highlights a lack of institutional capacity in Switzerland to address the growing and increasingly complex demands of HTA processes in the country. Addressing these deficiencies will require increased investment and resource allocation to ensure adequate support for HTA processes. In addition, the involvement of various agencies at the subnational level in producing HTA reports raises concerns about potential duplication of efforts and inefficiencies. Such challenges highlight the need for stronger coordination and the establishment of a unified HTA mechanism to harmonize processes, avoid redundancies, and more efficiently optimize the use of limited resources across federal and cantonal levels – a finding consistent with the view that an effective system requires balancing local autonomy with national coherence.

Additional findings on HTA implementation, capacity, and utilization indicated that the formal process for information gathering to support decision-making on new devices, drugs, and vaccines exists in Switzerland, with HTA being used mainly for reimbursement decisions and support of clinical guidance. These processes heavily rely on electronic databases, search engines, systematic reviews, and other evidence-based resources. The primary application areas where HTA was used as an element of measurement in the decision-making process included clinical interventions, medical devices, medicines, surgical interventions, and population-level health interventions. This broad scope reflects the versatility of HTA as a tool for evaluating diverse healthcare technologies. However, despite these established processes, converting information into practical policies continues to be a challenging endeavor. Respondents believed that emphasis on the advantages of HTA could assist in increasing political support and institutional capacity. Advocacy initiatives aimed at legislators and stakeholders could play a vital role in addressing these issues. In Switzerland, HTAs most frequently addressed safety, clinical effectiveness, and economic aspects, while social acceptability and feasibility were less commonly considered. Strong interest in expanding HTA capacity was expressed. However, significant impediments were pointed out, including limited awareness and advocacy for HTA, insufficient political support, limited institutional capacity, and persistent gaps in funding.

HTA policy is central in Switzerland. Among the most notable findings from this study are the policy inconsistencies governing HTA in Switzerland. While national-level policies exist, their implementation appears to be dispersed between regions and organizations, resulting in differences in how health technologies are evaluated and utilized. This decentralization presents advantages and disadvantages as it allows decisions to be tailored to local needs, yet it also perpetuates disparities and may delay the uptake of innovations. Switzerland could benefit from establishing a national HTA agency similar to those found in other INAHTA member countries [13]. Establishing a centralized agency would enable a more coordinated approach to HTA activities by reducing duplication of efforts and ensuring comprehensive coverage of health technologies. It could also serve as a platform for disseminating best practices and fostering stakeholder collaboration. Encouraging collaboration among government agencies, academic organizations, private-sector entities, and non-governmental organizations could help close existing gaps in expertise and resources. In the study, organizations and experts stated that increasing cooperation at both the national and subnational levels would improve the efficiency and comprehensiveness of HTA processes. Switzerland can promote greater buy-in and better facilitate the implementation of evidence-based recommendations by involving policymakers, healthcare providers, patients, and other relevant stakeholders throughout the assessment process.

Additionally, findings indicate that HTA policies and regulatory frameworks in Switzerland lack stability and consistency. Based on that, it is crucial to develop national action plans to strengthen and normalize these policies, thereby creating a more solid foundation. It is also important for the continuous application of HTA, as it was seen as applicable to maintaining the same level of understanding. It is also important to look at ways to use HTA for purposes other than making reimbursement choices. Broader application of HTA, such as wider health policy formulation and clinical practice guidance, could further increase its impact and visibility. By doing so, it could become a more integral part of healthcare governance in Switzerland. This can entail adding HTA to more general health policy formulation and clinical practice recommendations. This increases awareness of its importance and supports ongoing efforts to advocate for its role in healthcare decision-making. Effective collaboration and coordination are crucial and involve the input of professionals with diverse experience. It encourages collaboration and coordination among various HTA agencies at the national and sub-national levels, which makes the HTA system more cohesive and efficient.

Overall, the evidence from this study points to the need for a more balanced approach that enhances inter-cantonal coordination to better harmonize processes and reduce inefficiencies, while respecting the flexibility and local responsiveness that Switzerland's governance model offers. Switzerland should establish a national body to implement HTA. Although Switzerland has FOPH-led HTA processes, the absence of a fully integrated national approach and centralized system for coordinating and disseminating all HTA outcomes may lead to duplication and gaps in certain assessments.

According to the literature, alternative approaches could include bottom-up initiatives, with major hospitals acting as leaders in hospital-based HTA (HB-HTA) development and implementation [28]. More importantly, comparisons with other patterns observed in other health systems, as well as international experience, indicate that greater central coordination combined with regional participation can improve the consistency, uptake, and impact of evidence-based HTA recommendations [29]. For Switzerland, this balance could be met through a hybrid model that maintains local input while also implementing shared national standards via a well-coordinated mechanism or unified HTA agency. To ensure robust HTA governance and implementation, the coherence and integration of EU-level JCAs into Switzerland's national and canton-level HTA processes presents both opportunities and challenges. On one hand, JCAs provide a robust and standardized evidence base that can support Swiss HTA bodies, such as the FOPH and cantonal health authorities, in streamlining local assessments, reducing duplication of work, and accelerating decision-making. This is particularly valuable for smaller cantonal HTA offices or specialized committees that may have limited resources to conduct full clinical evaluations independently. On the other hand, Swiss HTA organizations must carefully consider national healthcare priorities, population characteristics, and the regulatory and reimbursement frameworks unique to Switzerland when interpreting and applying JCA findings. Adapting EU assessments to the Swiss context ensures that recommendations remain relevant and actionable, but may require additional methodological adjustments or supplementary analyses, such as incorporating local cost-effectiveness data or national clinical practice patterns [6,7]. Overall, leveraging JCAs can enhance efficiency and consistency in Swiss HTA while maintaining the flexibility necessary to address the specific needs of both federal and cantonal healthcare systems.

While this system analysis study provides a useful benchmark for understanding HTA in Switzerland, the expanded sampling that includes more Swiss Cantons is recommended to draw federal policy recommendations. Larger, more representative studies, including more organizations from each health sector, are needed for a comprehensive view of HTA in Switzerland. Given the country's size and the maturity of its HTA process, the limited number of organizations surveyed and interviewed may be considered a study limitation. This sample may not capture the full range of HTA experiences nationwide, and response proportions may not be generalizable to all of Switzerland.

## Conclusion

In conclusion, while Switzerland has made notable progress in implementing HTA processes, there are clear opportunities for improvement. This study identifies a good understanding of HTA among Swiss healthcare organizations, primarily in academic, public, and non-governmental sectors. Nevertheless, policy inconsistencies across Swiss Cantons/regions and organizations were also identified, which may lead to variances in how health technologies are assessed and utilized. Although Switzerland lacks a dedicated national HTA agency, a notable limitation that may contribute to gaps in technology assessment, the systematic procedures led by the Swiss FOPH provide a foundation for evidence-based decision-making.

To strengthen the system, Switzerland should consider several key improvements; establishing a national HTA agency to provide centralized oversight and coordination; standardizing policies and processes across Cantons/regions and organizations to ensure consistency in HTA implementation; fostering collaboration among diverse stakeholders; increasing financial investment and human resources dedicated to HTA; enhancing awareness among policymakers and healthcare providers; and expanding the use of HTA beyond reimbursement decisions. While local autonomy remains a strength of the Swiss governance system, its associated variations and duplication underscore the need for stronger inter-cantonal coordination. Such coordination, implemented while safeguarding regional flexibility, could be key in streamlining practices, improving efficiency, and promoting equitable access to health technologies.

## Supporting information

**S1 Text. HTA Questionnaire.**
(PDF)

**S2 Text. HTA Quali, Interview Guide, Text.**
(DOCX)

**S1 Checklist. COREQ Checklist.**
(DOCX)

## Acknowledgments

We acknowledge the support from Patience Vimbayi Mushamiri-Kuzviwanza and Maya Hassan with qualitative data analysis. We would also like to thank Rebecca Zucco for providing additional editing and proofreading support across all sections of the manuscript. Special thanks to Professor Marcel Tanner of the Swiss Tropical and Public Health Institute / R. Geigy Foundation, and Professor Matthias Schwenkglenks from the Faculty of Medicine, University of Basel, Basel, Switzerland, for their invaluable guidance and support throughout the study development, implementation, and review of the final manuscript. We also thank all experts who contributed to the review of the tools, and all participants who volunteered their time to participate in this study.

## Author contributions

**Conceptualization:** Mohammed Alkhaldi, Sara Ahmed.

**Data curation:** Mohammed Alkhaldi, Rima Kachach, Line Enjalbert, Aisha Al Basuoni.

**Formal analysis:** Mohammed Alkhaldi, Rima Kachach, Line Enjalbert, Aisha Al Basuoni, Malak Alrubaie.

**Funding acquisition:** Mohammed Alkhaldi, Sara Ahmed.

**Investigation:** Mohammed Alkhaldi, Aisha Al Basuoni.

**Methodology:** Mohammed Alkhaldi, Aisha Al Basuoni, Sara Ahmed.

**Project administration:** Mohammed Alkhaldi, Sara Ahmed.

**Resources:** Mohammed Alkhaldi, Aisha Al Basuoni, Sara Ahmed.

**Software:** Mohammed Alkhaldi.

**Supervision:** Mohammed Alkhaldi, Sara Ahmed.

**Validation:** Mohammed Alkhaldi, Line Enjalbert, Sara Ahmed.

**Visualization:** Mohammed Alkhaldi, Rima Kachach, Line Enjalbert.

**Writing – original draft:** Rima Kachach, Aisha Al Basuoni.

**Writing – review & editing:** Mohammed Alkhaldi, Rima Kachach, Line Enjalbert, Aisha Al Basuoni, Malak Alrubaie, Sara Ahmed.

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
