## [Decision Letter · Decision Letter 0]

23 Nov 2025

PGPH-D-25-02848

Health Technology Assessment System in Switzerland: Current State, Gaps, and Prospects for Improvement

Dear Dr. Alkhaldi,

Thank you for submitting your manuscript to PLOS Global Public Health. After careful consideration, we feel that it has merit but does not fully meet PLOS Global Public Health’s publication criteria as it currently stands. Therefore, we invite you to submit a revised version of the manuscript that addresses the points raised during the review process.

We look forward to receiving your revised manuscript.

Kind regards,

Paolo Angelo Cortesi, PhD

Academic Editor

Journal Requirements:

1. Please amend your detailed online Financial Disclosure statement. This is published with the article. It must therefore be completed in full sentences and contain the exact wording you wish to be published.

a) State the initials, alongside each funding source, of each author to receive each grant. For example: “This work was supported by the National Institutes of Health (####### to AM; ###### to CJ) and the National Science Foundation (###### to AM).”

For more information, please go to our submission guidelines:

https://journals.plos.org/globalpublichealth/s/submission-guidelines#loc-financial-disclosure-statement

2. Please ensure that the funders and grant numbers match between the Financial Disclosure field and the Funding Information tab in your submission form. Note that the funders must be provided in the same order in both places as well.

3. Please update your online Competing Interests statement. If you have no competing interests to declare, please state: “The authors have declared that no competing interests exist.”

4. In the online submission form, you indicated that “Anonymized data used in the current study are available upon request from the authors. Anonymized data used in the current study are available upon request from the authors.”.

a) In a public repository,

b) Within the manuscript itself, or

c) Uploaded as supplementary information.

For further assistance, you may go to: http://journals.plos.org/globalpublichealth/s/data-availability

5. Please provide separate main figure files in .tif or .eps format only and ensure that all files are under our size limit of 10MB.

6. We have noticed that you have a list of Supporting Information legends in your manuscript. However, there are no corresponding files uploaded to the submission. Please upload them as separate files with the item type 'Supporting Information'.

Additional Editor Comments (if provided):

Reviewers' comments:

Reviewer's Responses to Questions

**Comments to the Author**

1. Does this manuscript meet PLOS Global Public Health’s publication criteria?

Reviewer #1: Yes

Reviewer #2: Partly

2. Has the statistical analysis been performed appropriately and rigorously?

Reviewer #1: I don't know

Reviewer #2: N/A

3. Have the authors made all data underlying the findings in their manuscript fully available (please refer to the Data Availability Statement at the start of the manuscript PDF file)?

Reviewer #1: Yes

Reviewer #2: No

4. Is the manuscript presented in an intelligible fashion and written in standard English?

Reviewer #1: Yes

Reviewer #2: Yes

Reviewer #1: General: This study is very timely as with the HTAR starting its implementation in 2025, though EEA countries are not part of the Coordinating Group, they will need to continue to be part of the evolving HTA landscape in Europe and I think therefore a review of the HTA system is very helpful for the country. That said it maybe helpful to mention the HTAR in the manuscript in the abstract and definitely in the introduction otherwise this is looking very dated as the landscape some 2-3 years back. Seeing the dates of the research – this is clear that it is some years old, but then the context and introduction & discussion should reflect the HTA scenario in 2025.

https://health.ec.europa.eu/health-technology-assessment/implementation-regulation-health-technology-assessment_en

I think the study is very good but needs better and improved reporting and putting into context for it to be really valuable and helpful.

Introduction:

• EunetHTA initially started in 2006 as a framework 7 EU funded project 2006-2008 then there was a transition period for a year and then as you say Joint Action 1 was from 2010. So currently this isn’t clear – or reads incorrect, as I worked for EunetHTA on behalf of UK from 2008, I think this needs to be rectified. https://www.healthinformationportal.eu/european-initiative/european-network-health-technology-assessment

• EuneHTA21 also officially shut down in September 2023 in anticipation of the HTAR – so the tense needs to be changed e.g. “The EUnetHTA is composed of 35

government-appointed organizations from 24…” it should say “EUnetHTA was composed of …”

• The sentence here is again not a 1005 clear “Although there is some standardization in national HTA systems across Europe, notable differences remain in both process and methodology” With the HTAR now there is joint clinical assessment – other domains economic, other considerations etc and actual appraisal or decision making at done differently in different countries but it is important to clarify that 27 EU countries now have to follow the same process for clinical assessments and have joint clinical assessments for oncology, ATMPs already with selected medical devices soon starting then orphan drugs in 2028 and everything from 2030 – this is very important and is missing from the context.

• There is also some strange phrasing – e.g. “However, the statutory health insurance system, including the benefit package, is managed centrally at the federal level. Switzerland has the second-highest life expectancy (82.8 years) in Europe, and healthy life expectancy exceeds the EU average by several years” – how and where the benefit package is managed shouldn’t be in contrast with the life expectancy – so the use of however seems strange.

• Clearly the insurance and reimbursement system is fragmented and complex in Switzerland so I appreciate the authors trying to describe it – however it isn’t completely clear what is done by national level and what by cantons and how they interplay – maybe a figure could be helpful here.

Methods

• The HTA survey was administered to 9 organizations of HTA in the country but the total number of organizations working in HTA in the country is unclear – ie. How many are there in Switzerland so the reader can understand how representative this sample is so please include. Additionally later it talks about 13 institutions – please clarify the numbers.

• Also to confirm there were 8 in depth interviews with 9 experts – does that mean 2 were done together, were they from the same organization? Please clarify.

• Otherwise, the methods sound robust – when WHO also does a situational analysis of HTA in a country we do a literature review, use the survey as well as do in-depth interviews to get a complete picture.

Results

• These appear to be comprehensive

Discussion

• This would benefit with some restructuring and reframing – maybe highlight the discussion elements under some sub headings for clarity of what you are trying to highlight, or focus on.

• Additionally, once again maybe putting into the context of HTAR – highlighting how the JCAs coming from the EU level could be considered by national or canton level organizations.

Reviewer #2: This is a small preliminary study regarding the HTA landscape in Switzerland. The authors note this limitation. I would be interested to know why the authors chose to conduct this study. It is important to spread the word about the importance of HTA in decision-making however I question what this article will provide for others who may be setting up HTA in their country, as there are always different contexts. I didn’t find the article particularly well written (although acceptable) and some statements were made without any reference to evidence e.g. the statement about registries and their limited quality and utility. There was a process described for identifying organisations to survey but it really wasnt very clear e.g. research members talking to collaborators they knew in Switzerland.

**Do you want your identity to be public for this peer review?** For information about this choice, including consent withdrawal, please see our Privacy Policy

Reviewer #1: **Yes:** Tarang Sharma

Reviewer #2: No

---

## [Decision Letter · Decision Letter 1]

4 Jan 2026

Health Technology Assessment System in Switzerland: Current State, Gaps, and Prospects for Improvement

PGPH-D-25-02848R1

Dear Dr. Alkhaldi,

We are pleased to inform you that your manuscript 'Health Technology Assessment System in Switzerland: Current State, Gaps, and Prospects for Improvement' has been provisionally accepted for publication in PLOS Global Public Health.

Best regards,

Paolo Angelo Cortesi, PhD

Academic Editor

Reviewer Comments (if any, and for reference):

Reviewer's Responses to Questions

**Comments to the Author**

Reviewer #2: All comments have been addressed

publication criteria?

Reviewer #2: Yes

3. Has the statistical analysis been performed appropriately and rigorously?

Reviewer #2: N/A

4. Have the authors made all data underlying the findings in their manuscript fully available (please refer to the Data Availability Statement at the start of the manuscript PDF file)?

Reviewer #2: Yes

5. Is the manuscript presented in an intelligible fashion and written in standard English?

Reviewer #2: Yes

Reviewer #2: The manuscript has been substantially improved based on the reviewers comments. My only comment is about Figure 3 that seems to be missing the designation for dark blue in the legend.

**Do you want your identity to be public for this peer review?** For information about this choice, including consent withdrawal, please see our Privacy Policy

Reviewer #2: No
